# The Effect of the COVID-19 Pandemic on Early Adolescent Fractures in the Republic of Korea

**DOI:** 10.3390/medicina59091671

**Published:** 2023-09-15

**Authors:** HyunSeo Park, Hogyeong Kang, Siyeong Yoon, Simho Jeong, Soonchul Lee

**Affiliations:** 1CHA University School of Medicine, 120 Hyeryong-ro, Pocheon-si 11160, Gyeonggi-do, Republic of Korea; wiseclue@chauniv.ac.kr (H.P.); hksteven@chauniv.ac.kr (H.K.); 2Department of Orthopaedic Surgery, CHA Bundang Medical Center, CHA University School of Medicine, 335 Pangyo-ro, Bundang-gu, Seongnam-si 13488, Gyeonggi-do, Republic of Korea; tldud1105@naver.com (S.Y.); gmsim6727@gmail.com (S.J.)

**Keywords:** COVID-19, orthopedics, pediatrics, fractures, adolescents

## Abstract

*Background and Objectives:* Restrictions on daily activities to slow down the propagation of COVID-19 have changed the epidemiological pattern of pediatric fractures in many countries. However, the effect of the pandemic on pediatric fractures has not been fully studied. In this study, we investigated the impact of COVID-19 on early adolescent fractures in Korea. *Materials and methods:* We conducted a retrospective follow-up on a nationwide cohort of Korean early adolescents born between 2006 and 2009. The prevalence and incidence of pediatric fractures and the frequency of surgical treatment were compared between two different eras. *Results:* The prevalence and incidence of fractures during the pandemic have both shown a significant decrease: prevalence reduced from 34,626 to 24,789 (*p* < 0.001), while incidence decreased from 29,804 to 18,898 (*p* < 0.001). Considering sex, the shift in fracture prevalence was statistically significant (*p* = 0.020), whereas the incidence was not (*p* = 0.862). The decline in both fracture prevalence and incidence exhibited significant variation across birth year groups (prevalence, *p* < 0.001; incidence, *p* < 0.001), with a more pronounced reduction observed in the older age groups. While the proportion of patients who required surgeries has increased, the mean frequency of surgical treatment per patient remained at a similar level (by prevalence, *p* = 0.181; by incidence, *p* = 0.735). The decline in both fracture prevalence and incidence has shown significant variation in relation to fracture sites (prevalence, *p* < 0.001; incidence, *p* < 0.001), with a decrease in distal limb fractures and an increase in forearm and axial body fractures. *Conclusions:* The pediatric fracture pattern in Korea has been notably influenced by the COVID-19 pandemic, warranting further investigation into causal factors. Our findings should help predict epidemiology in the post-pandemic period and thus aid policymaking and patient management.

## 1. Introduction

The first case caused by the novel coronavirus (SARS-CoV-2) was reported on 31 December 2019 [1]. The virus soon spread internationally, and the World Health Organization (WHO) declared a global health emergency on 30 January 2020 [2] and a global pandemic on 11 March 2020 [3]. In the Republic of Korea (hereafter South Korea), the first case of COVID-19 was confirmed on 20 January 2020, about 20 days after the outbreak started in China. Although it was not deemed significant at that time, the number of confirmed cases suddenly increased exponentially beginning on 22 February, with the cumulative number of COVID-19 patients reported accordingly: 422 on 22 February, 1111 on 25 February, 2728 on 28 February, 4272 on 2 March, 6738 on 6 March, and 8271 on 16 March. The government elevated the level of alert from “orange” (cautious) to “red” (serious) on 23 February [4].

To mitigate the transmission of COVID-19, many countries have enforced infection control measures, including canceling public gatherings and mass closures of childcare facilities, schools, and workplaces [5,6,7]. In South Korea, the universal use of face masks was mandated in February 2020, and social distancing schemes with a range of restriction levels were instituted a month later. The Korean government, however, mainly used non-pharmaceutical interventions, such as suggesting voluntary work-from-home conditions and local quarantine around confirmed patients, instead of implementing a strict lockdown [8]. Thanks to the strengthening of social distancing and growing interest in personal hygiene, the steep increase in infection cases began to subside in mid-March [9].

Social distancing and lockdown measures have brought drastic lifestyle changes. By the nature of “shelter-in-place” orders and “stay-at-home” policies, time spent in a home setting has increased while outdoor activities have reduced during the COVID-19 pandemic, and such change to a sedentary lifestyle has led to health problems. A survey study in Korea reported that 96.7% and 83.4% of the respondents avoided outdoor activities and crowded places in the early phase of the pandemic, respectively [10], and another study revealed that the duration of moderate- to high-intensity physical activity has significantly decreased after the outbreak in Korea [11]. Similar changes were seen in other parts of the world. A study of 35 research organizations from Europe, North Africa, Western Asia, and the Americas, shows that daily sitting increased by 60% [12]. The “new normal” lifestyle has also led to a negative impact on human psychology: people experienced symptoms of distress, depression, post-traumatic stress disorder, anxiety, frustration, and suicide [13]. The vulnerable populations are no exceptions for suffering unintended secondary health consequences; in fact, they have been experiencing even greater impacts due to reduced contact with caretakers and healthcare providers [14,15].

As it seems natural that restrictions on physical activities would affect the musculoskeletal system, a number of studies have investigated the impact of the COVID-19 pandemic in the orthopedic field. It has been shown that the incidence of orthopedic diseases, including fractures of various sites, decreased during the pandemic, especially at the beginning of the outbreak, although the extent of change differed [14,15,16,17]. Many studies have also shown that the volume of surgical treatment has decreased during this period, as elective surgeries were delayed [17,18]. Some later studies have shown that the total volume of orthopedic surgery sprung back as COVID-19 continued; however, this is mostly because of the surgeries that could not be postponed any longer [19].

Fracture in children is a significant issue in public health since various complications such as traumatic arthritis, ischemic osteonecrosis, and osteofascial compartment syndrome may develop. These diseases and/or conditions can cause not only physical pain but also pediatric limb deformities [20]. Hence, it is essential to understand the pattern of childhood fractures and utilize the knowledge to minimize future occurrences.

Many studies have shown that the decline in the number of total pediatric visits and the number of pediatric and adolescent fractures after the emergence of COVID-19 is a global trend. One comprehensive meta-analysis found a substantial 43% decline in the incidence of fractures present at hospitals compared to the pre-pandemic period [15]. Further bolstering this evidence, a study conducted in China illuminated the impact of home confinement during the COVID-19 pandemic, highlighting a significant reduction in pediatric fractures [20]. Similarly, another study in China echoed these trends, indicating a noteworthy decrease in pediatric hospital visits, with daily numbers dwindling to approximately one-quarter of historical data from 2019 [21]. Beyond China, research from diverse regions, including Korea and France, converged on a shared observation, revealing a substantial reduction in the overall volume of pediatric emergency department visits during the COVID-19 pandemic [22,23]. These findings collectively underscore the far-reaching impact of the pandemic on pediatric healthcare utilization and fracture incidences worldwide.

However, to our knowledge, there is a lack of investigations on how the epidemiology of early adolescent fractures has changed during the pandemic on a nationwide scale. We hypothesized that the incidence and prevalence of a wide range of fractures in an adolescent group might have changed during the pandemic era. Therefore, we followed up with a large-scale, population-based, nationwide cohort in South Korea to observe any changes in the epidemiological pattern of fractures after the emergence of the COVID-19 pandemic.

## 2. Materials and Methods

We conducted a retrospective nationwide cohort study with data from the Korean National Health Insurance Service (NHIS) in South Korea. All Korean nationals (e.g., 51,780,000 people in 2019) are obliged to get mandatory national health insurance, enabling NHIS to collect data on a nationwide scale. The claim-based NHIS database used in this study included all the medical records between January 2006 and December 2020 about the population who was born between 2006 and 2009. This extensive database provided information about the prevalence and incidence of patients with fractures and those who underwent surgery after fractures. We also collected demographic information including age, sex, and site of fractures. Fractures were classified according to the International Classification of Disease, Tenth Revision (ICD-10) codes, as recorded in the NHIS database. We obtained data on fractures on every site except for skull and facial bones (S02): axial fractures (S12, S22, S32), upper extremity fractures (S42, S52, S62), and lower extremity fractures (S72, S82, S92).

There were no exclusion criteria used in this study, as the NHIS covers the whole population, as mentioned above, and includes all record data from every primary clinic and tertiary hospital in South Korea. Furthermore, each patient has their own registration number and, therefore, is uniquely identified in the database. This means that there was no duplication of the diagnoses and/or patient identification. 

To compare the abovementioned prevalence and incidence between pre-COVID-19 and COVID-19 eras, data for 2019 and 2020 were analyzed, respectively, based on the criteria of age, sex, and site of fractures. Consequently, early adolescents (aged 10–13 in 2019 or born in 2006–2009) were analyzed as our study population. The cohort constituted 1,117,236 and 1,033,991 individuals representing 2019 and 2020, respectively. For the incidence study, we considered the date of the earliest claim with a registration code as the incident time, and the patient as an incident case in that year. A flow diagram describing the prevalence and incidence of patients with fractures and patients who underwent surgery after fractures is shown in Figure 1. 

The basic characteristics in this study were presented as frequencies and percentages, with mean values and standard deviations calculated. For the statistical analysis, Pearson’s chi-square test and an unpaired t-test were utilized to compare categorical and continuous pediatric fracture characteristics between the two years, respectively. A *p*-value less than 0.05 was considered statistically significant. Regarding the subgroup analyses, we divided the participants by sex and birth year. All statistical analyses were conducted using R Statistical Software version 3.6.3 (The R Foundation for Statistical Computing, Vienna, Austria).

## 3. Results

### 3.1. Demographic Pattern

Table 1 shows the basic characteristics of the study population. The prevalence of fractures during the pre-pandemic year was 34,626/1,117,236 (3.10%), and the incidence was 29,804/1,117,236 (2.67%). In 2020, the former decreased to 24,789/1,022,991 (2.42%) and the latter to 18,898/1,022,991 (1.83%) (prevalence, *p* < 0.001; incidence, *p* < 0.001).

In both years, the number of male patients was almost twice the number of female patients. In the pre-COVID-19 era (2019), the prevalence of fractures among male and female patients stood at 23,179 and 11,447, respectively, whereas in the COVID-19 era (2020), these figures were 16,819 for males and 7970 for females. In terms of incidence, 2019 reported 19,784 fractures for males and 10,020 for females, whereas 2020 witnessed 12,559 male fractures and 6339 female fractures. A notable distinction emerged in the prevalence ratio between the two genders (*p* = 0.020), with an increase from 66.9% to 67.8% in male patient fracture prevalence. However, the incidence of fractures by gender remained unchanged and statistically non-significant (*p* = 0.862). 

The study population was subdivided into four groups by the year they were born, resulting in four age groups: 10, 11, 12, and 13, according to their age in 2019. In the pre-COVID-19 era (2019), both prevalence and incidence exhibited an upward trend from age 10 to 13, while in the COVID-19 era, both prevalence and incidence reached their highest point in the age 12 group. The number of fractures decreased in all four groups. Significantly, the degree of reduction in both fracture prevalence and incidence varied considerably across age groups (prevalence, *p* < 0.001; incidence, *p* < 0.001), with a slight increase observed in the proportions of the younger two groups during the pandemic. 

### 3.2. Proportion of Surgical Management

Table 2 shows the number of fracture patients who received operative treatment and the mean number of surgeries that such a patient received. In the given period, the prevalence of surgical treatment decreased from 2870 to 2296 and incidence from 2341 to 1676. This means that the percentage of patients who required surgery after a fracture increased from 8.29% to 9.26% and from 8.15% to 9.13%. The frequency of surgeries, however, remained at a similar level (mean difference with a 95% confidence interval by prevalence, 0.021 (−0.010, 0.052); by incidence, 0.006 (−0.029, 0.041)).

### 3.3. Fracture Sites

Table 3 presents the prevalence and incidence of pediatric fractures during 2020 (COVID-19 era) and the reference period 2019 (pre-COVID-19 era), by fracture site. The most common fracture sites were, in order of frequency, the foot, lower leg, including the ankle, forearm, and lumbar spine and pelvis, in both 2019 and 2020. Although the absolute frequency of pediatric fractures by fracture site showed a decreasing tendency, the extent of this decline varied significantly across different fracture sites (prevalence, *p* < 0.001; incidence, *p* < 0.001). Notably, the prevalence and incidence of fractures in the distal part of limbs (fractures of the foot, lower leg, including the ankle, and the wrist and hand level) declined in percentage, whereas fractures in the forearm and axial part of the body (fractures of the lumbar spine, pelvis, ribs, sternum, and thoracic spine) increased in percentage.

## 4. Discussion

It is not surprising that the number of early adolescent fractures declined dramatically during the COVID-19 pandemic, as many studies have already reported an average decline of 43% (35–50%) in the number of fractures during the pandemic compared to pre-pandemic periods [9]. The reduction rate of the incidence of fractures in early adolescence (36.6%) was higher than the overall underage population throughout South Korea over the same period (29.25%) [22]. Previous studies have also shown greater reductions of fractures in adolescents compared to the whole pediatric population, which results in declined mean ages of the patients. A widely accepted explanation for the tendency is that the restrictions had a greater impact on the children, as the incidence of physical activity-related fractures increases with age. The smaller activity space of younger children, including infants and preschool children, also accounts for the trend. Their daily activities are not as impacted by restrictions as older children [20,24,25,26]. Notably, the reduction rate of the early adolescent fracture numbers was smaller than in previous studies performed in other countries [27,28]. This can be explained that the limited social distancing program carried out in South Korea compared to strict lockdown policies in other countries, which had a smaller impact on the restriction of physical activity, and thus fracture events.

It is known that boys are much more likely to suffer fractures than girls. Although the percentage difference varies by region and period, the male proportions have been often reported to exceed 60% [29,30]. Most scholars believe that this is due to the relatively higher-level activities of boys and their interest in taking risks, which may lead to the onset of fractures [31,32,33]. Some studies have shown that infection control measures have reduced the male-to-female ratio, stating that the restrictions have resulted in more powerful fracture prevention in boys, who participate in sports activities more frequently [20,27,34,35,36]. Other studies, on the other hand, have reported that there was no significant change in the sex distribution, which corresponds to our result. The lack of variance suggests that the restriction measures have equally affected the populations throughout the study cohort, aligning with our study’s finding of no significant alteration in the gender distribution of incidence [22,26,37,38]. To the best of our knowledge, there have been no reports regarding changes in the sex ratio of pediatric fracture prevalence. The observed shift in the sex ratio of prevalence in this study, though modest at less than 1%, is still statistically significant. This shift could potentially be attributed to the higher severity of fractures in males, coupled with delayed recovery and reduced healing conditions resulting from limited access to medical care during the COVID-19 pandemic era.

Our study has shown that the mean number of surgical treatments each patient received remained at a similar level between the pre-pandemic and pandemic years. Studies on how the COVID-19 pandemic has affected the percentage of operative management in fracture cases had conflicting results. Some reported insignificant changes or decreases in the ratio of surgical treatments [27,35]. It is intriguing to note that many studies, especially those that investigated pediatric populations, have shown that surgical treatments have decreased in numbers but increased in percentage, which is similar to our study [28,35,39,40]. It is speculated that such discrepancies arose from two factors. One is the difference in the necessity of surgical management in different age groups. Although the pandemic has imposed delays on elective surgeries, emergency surgeries were performed as needed. It has been shown that the number of elective surgeries has significantly decreased during the pandemic, while trauma and emergency surgeries have not decreased [16]. Since surgical treatments for fractures are more often required in younger patients, the number of surgeries has decreased less in children [41,42]. We assume that this, combined with the reluctance to visit hospitals for “light” injuries during the pandemic, has led to an increase in the surgery ratio of early adolescent fractures. The other is the time range of the data collected. The reduced volume of orthopedic surgeries due to the COVID-19 pandemic has gradually recovered over time [16]. As the data collected in the earlier studies reflect shorter periods at the beginning of the pandemic, the decrease in the number of surgical treatments might have been exaggerated when compared to later studies, which include the “recovery” period of operations. 

Some prior studies demonstrated controversial changes in the relative prevalence and incidence of pediatric fractures by fracture sites; for example, some reported an increase in upper limb fractures whereas others reported a decrease [24,26,34,36,40]. We assume this inconsistency among studies, including ours, is because of the ignorance of other parameters, such as region, race, and climate.

Our study has revealed significant changes in pediatric fracture site patterns before and during the COVID-19 pandemic. Although our primary data do not include information regarding the injury mechanism, our results imply the mechanism of pediatric fracture has changed during the pandemic. The proportion of fractures on the distal part of the limbs (i.e., foot, lower leg and ankle, wrist, and hand) decreased during the COVID-19 pandemic. As the distal extremities are frequently used in sports activities, a probable cause of this change could be a decrease in activities due to social distancing and lockdown policies. Coherently, prior studies pointed out the significant decrease in sports-related injuries during the COVID-19 pandemic [19,24,35]. On the other hand, the relative prevalence and incidence of fractures in the axial part of the body, including the spine, pelvis, ribs, and sternum, increased during the pandemic slightly. We paid attention to the above-mentioned sites, which are common and specific fracture sites indicating relatively higher energy trauma, such as car accidents.

We believe our research on changes in pediatric fracture epidemiological patterns during the COVID-19 pandemic is clinically important for several reasons. From the shifts in pediatric fracture distribution, healthcare providers could be informed about the specific risks and vulnerabilities of pediatric patients during the pandemic. Furthermore, identifying shifts in fracture patterns can help clinicians prioritize and adapt their treatment strategies to address the changing needs of pediatric patients. For example, as fractures in distal extremities, which often result from outdoor activities, have significantly decreased during the pandemic, healthcare providers can prioritize injury prevention education pertaining to reduced physical activity or accidents within the home environment. The implementation of unprecedented social distancing measures and lockdowns during the COVID-19 pandemic garnered both support and criticism. Recognizing the potential for future pandemics necessitating similar measures, it becomes imperative to conduct longitudinal studies to analyze their impact on pediatric fractures. Such research assumes a pivotal role in informing the development of enduring and targeted public health strategies and interventions specifically tailored to mitigate the occurrence of pediatric fractures in the context of future outbreaks. Overall, understanding changes in pediatric fracture patterns during the pandemic can lead to improved clinical care, injury prevention strategies, and public health policies that prioritize the unique needs of pediatric patients in challenging circumstances.

This study is the first to compare the epidemiology of early adolescent fractures during the COVID-19 pandemic compared to before the pandemic on a nationwide scale. Using NHIS data, which covers the entire Korean population without exception, statistical power was obtained. Also, we obtained diagnosis records on all types of fractures, except for skull and facial fractures, and compared the change in fracture patterns by site. 

However, there are several limitations to consider in our study. As mentioned, our study group only included early adolescents due to the limited data. Also, the NHIS database solely includes data on the Korean population, which is known for its monoethnicity. As the epidemiology of pediatric fractures is influenced by region, ethnicity, and climate, this research would not fully reflect the global epidemiological changes in pediatric fractures. Furthermore, we could not identify the mechanism of injury in our data. As lifestyle has changed drastically due to COVID-19, we believe that the mechanism of pediatric fractures shows differences before and during the pandemic. Going along with our assumption, one study showed a significant association between home confinement during the COVID-19 outbreak and a reduction in pediatric fractures [20]. Therefore, further studies should investigate the relationship between the possible change in the mechanism of pediatric fractures and the COVID-19 outbreak.

## 5. Conclusions

In this study of early adolescents born between 2006 and 2009 in South Korea, we observed a significant decrease in pediatric fractures during the COVID-19 pandemic, with notable gender differences, as the prevalence increased among males but decreased among females. The extent of this decline varied by age group, showing a slight increase in the younger two groups. Furthermore, we noted shifts in fracture sites, with an increase in fractures in the distal part of limbs and a decrease in the forearm and axial body, underscoring the substantial impact of the COVID-19 pandemic on pediatric fractures and highlighting the importance of ongoing preparedness efforts. With collective efforts to recognize how fracture patterns have altered during the pandemic, healthcare providers would be able to adjust treatment approaches and enhance overall care, injury prevention measures, and public health policies, with a focus on addressing the specific needs of pediatric patients in such demanding situations.

## Figures and Tables

**Figure 1 medicina-59-01671-f001:**
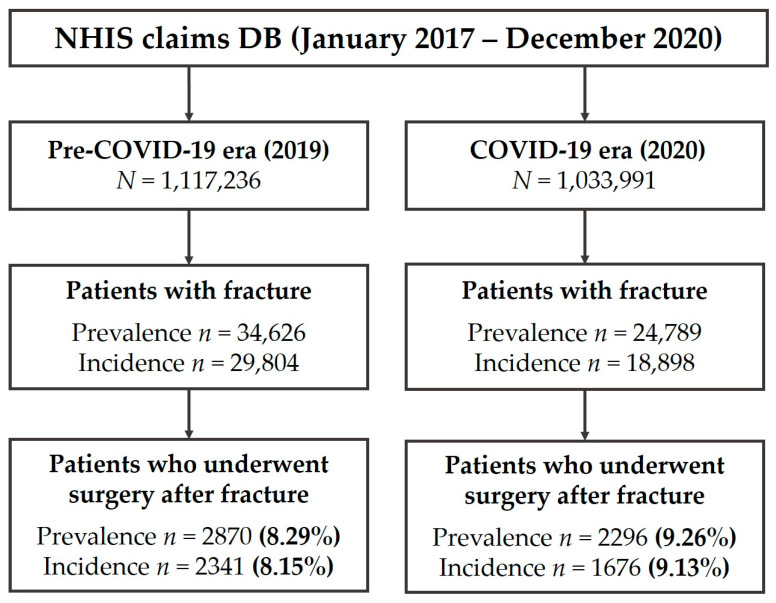
A flow diagram of patients with fractures from the Korean nationwide cohort before and during the COVID-19 pandemic.

**Table 1 medicina-59-01671-t001:** Basic characteristics of the study population in 2019 and 2020.

		Fracture Prevalence			Fracture Incidence		
		Pre-COVID-19 Era (2019) (*N* = 1,117,236)	COVID-19 Era (2020) (*N* = 1,033,991)	*p*-Value ^1^	Pre-COVID-19 Era (2019) (*N* = 1,117,236)	COVID-19 Era (2020) (*N* = 1,033,991)	*p*-Value ^1^
Sex	Male	23,179 (66.9%)	16,819 (67.8%)	0.020	19,784 (66.4%)	12,559 (66.5%)	0.862
	Female	11,447 (33.1%)	7970 (32.2%)		10,020 (33.6%)	6339 (33.5%)	
Age ^2^	10	3530 (10.2%)	3112 (12.6%)	<0.001	3091 (10.4%)	2539 (13.4%)	<0.001
	11	8489 (24.5%)	6805 (27.5%)		7384 (24.8%)	5360 (28.4%)	
	12	10,801 (31.2%)	7710 (31.1%)		9318 (31.3%)	5854 (31.0%)	
	13	11,806 (34.1%)	7162 (28.9%)		10,011 (33.6%)	5145 (27.2%)	
Total		34,626	24,789	<0.001	29,804	18.898	<0.001

^1^ Pearson’s chi-square test. ^2^ Age in 2019.

**Table 2 medicina-59-01671-t002:** Frequency of surgical treatment per patient.

	Pre-COVID-19 Era (2019) (*N* = 1,117,236)	COVID-19 Era (2020) (*N* = 1,033,991)	Mean Difference (95% Confidence Interval) ^1^	*p*-Value ¹
Prevalence	2870 (8.29%)	2296 (9.26%)		
Mean ^2^ (SD)	1.223 (0.543)	1.244 (0.581)	0.021 (−0.010, 0.052)	0.184
Incidence	2341 (8.15%)	1676 (9.13%)		
Mean ^2^ (SD)	1.234 (0.557)	1.240 (0.551)	0.006 (−0.029, 0.041)	0.735

^1^ Unpaired t-test. ^2^ Frequency of surgeries per patient.

**Table 3 medicina-59-01671-t003:** Prevalence and incidence of pediatric fractures by fracture site.

		Fracture Prevalence			Fracture Incidence		
ICD-10 Code	Fracture Site	Pre-COVID-19 Era (2019)	COVID-19 Era (2020)	*p*-Value ^1^	Pre-COVID-19 Era (2019)	COVID-19 Era (2020)	*p*-Value ^1^
S12	Neck	13 (0.04%)	7 (0.02%)	<0.001	11 (0.04%)	7 (0.04%)	<0.001
S22	Rib(s), sternum and thoracic spine	12 (0.03%)	17 (0.06%)		12 (0.05%)	12 (0.07%)	
S32	Lumbar spine and pelvis	1327 (3.83%)	1195 (4.16%)		1207 (4.87%)	968 (5.27%)	
S42	Shoulder and upper arm	482 (1.39%)	363 (1.26%)		390 (1.57%)	249 (1.36%)	
S52	Forearm	3216 (9.29%)	2841 (9.89%)		2626 (10.59%)	2047 (11.15%)	
S62	Wrist and hand level	1108 (3.20%)	843 (2.93%)		990 (3.99%)	648 (3.53%)	
S72	Femur	215 (0.62%)	200 (0.70%)		125 (0.50%)	91 (0.50%)	
S82	Lower leg, including ankle	6671 (19.26%)	3987 (13.88%)		5495 (22.17%)	2811 (15.32%)	
S92	Foot, except ankle	24,157 (69.76%)	17,121 (59.61%)		21,098 (85.11%)	13,370 (72.85%)	
Sum ^2^		37,201	26,574		31,954	20,203	

^1^ Pearson’s chi-square test. ^2^ Percentages may add to more than 100% as a result of patients being given multiple diagnosis codes.

## Data Availability

Not applicable.

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
