# Peer review of "The Effect of the COVID-19 Pandemic on Early Adolescent Fractures in the Republic of Korea"

_medicina, 2023, doi:10.3390/medicina59091671_

Round 1

Reviewer 1 Report

Dear Authors: This is an interesting concept and very original. Thanks!   Please:   1. A clear abstract will improve the paper. 2. - Abstract and Introduction - The statement of aim is very vague and needs reframing. What is the main question addressed by the research?  3. The Abstract needs to include a more sepcific Conclusion. 4. Introduction: Summarizes recent research related to the topic. Insert recent literature and add more “outputs”. Establishes the originality of the study. 5. Line 80: please, insert IBM® SPSS Statistics. 6. Results: Examine the data tables to see if there are statistically significant differences. Do the data support the conclusions? 7. The Discussion needs to be reduced and revised to compare and contrast findings with other. They should make reference to statistical analyses. The outcome should be a critical analysis of the data “collected”. Authors should describe and discuss the overall “story” formed and future research might confirm the findings. 8. The conclusion needs to be rewritten as 1-2 clear concise sentences which state the key finding and should reflect upon the aim. 9. References: should be relevant and readily retrievable. 10. A professional editor needs to edit the entire manscript.   Thanks! Kind Regards 

Moderate editing of English language required.

Reviewer 2 Report

I appreciate the opportunity to review this manuscript, the article is well written and may be of interest to medical teams dealing with adolescent fractures. However, some concerns that I detail below should be addressed by the authors to improve the quality of their manuscript before considering publication

1) The author reported that the number of fractures has decreased a lot "after" the outbreak of the pandemic. However, they only collected data up to 2020. Therefore, the outbreaks in 2021 (e.g. Omicron subvariants BA.4 and BA.5) were not taken into account. Therefore, I suggest changing the term "after" to "during" throughout the manuscript.

2)  Please provide quantitative data in your abstract, this information would be valuable for the readers of your manuscript.

4) Please provide further substantiation to support your hypothesis that the number of fractures decreased after the outbreak of the pandemic in a group of adolescents.

5) Please indicate the software you used to perform your statistical analysis.

6) Elaborate on the clinical implications of your finding and the relevance and decision making in the discussion of your manuscript.

7) The authors conclude that "The number of fractures in our study cohort of early-adolescents in South Korea decreased after the emergence of COVID-19 while the rate of surgery increased". However, they do not present quantitative results in the results section to support this judgement. Please provide the effect size (mean difference), 95% confidence interval and p-value, for comparisons (i.e. paired t-test).

Minor editing of English language required

Reviewer 3 Report

The research aim is to analyze the effect of COVID-19 pandemic on the epidemiological patterns and to predict the post-pandemic volumes of fractures in adolescents.

Minor editing error – the corresponding author should be noted with a different sign from the author who contributed equally to the research.

The abstract is structured appropriately.  

The introduction transposes the research into the topic and formulates the objective of the study at the end. However, I suggest extending this section as it is too short. More information about influence of COVID-19 on the orthopedic field should be added in relation to other scientific papers, for e.g. Moldovan, F.; Gligor, A.; Moldovan, L.; Bataga, T. An Investigation for Future Practice of Elective Hip and Knee Arthroplasties during COVID-19 in Romania. Medicina 2023, 59, 314. doi: 10.3390/medicina59020314

In the methodology section the statistical analysis should be better described. What software was used? Where there no normality distribution tests performed?

In the results section no statistical tests were performed for the demographic pattern and fracture sites. The authors present just the descriptive statistics. Table 1 should be revised - Sex and Age column should pe written only once in the left of the table.

The discussions interpret the research results and relate them to other results from the scientific literature. Limitations of the study are presented.

The conclusions are concise and clear.

The references are adequate but need proper editing according to the journal criteria and can be extended as suggested above.

Round 2

Reviewer 1 Report

Dear Authors

Thanks!

Kind regards

-

Author Response

Dear Reviewer,

We appreciate your consideration.

Best regards,

Reviewer 2 Report

I thank the authors for correctly addressing each of my comments. The quality of the manuscript has improved markedly. There is only two points that was suggested that was not fully responded to.

1) I believe it would be relevant to report in Table 2, the mean difference and 95% confidence interval for the Unpaired t-test, since the p-value is usually not sufficient for this type of comparison.

2) The clinical relevance of their results is now presented in a more valuable way. Sentences "identifying shifts in fracture patterns can help clinicians prioritize and adapt their treatment strategies to address the changing needs of pediatric patients" and "understanding changes in pediatric fracture patterns during the pandemic can lead to improved clinical care, injury prevention strategies, and public health policies that prioritize the unique needs of pediatric patientsin challenging circumstances" are very relevant and it would be interesting to highlight in the conclusions a short sentence summarizing this message. 

Author Response

Dear Reviewer,

According to your valuable feedback regarding enhancing the statistical robustness of our analysis comparing the frequency of surgical treatments before and during the COVID-19 pandemic, we have computed the mean difference with a 95% confidence interval.

In the conclusion section, we have incorporated an additional sentence to address the clinical significance of our findings: “With collective efforts to recognize how fracture patterns have altered during the pandemic, healthcare providers would be able to adjust treatment approaches and enhance overall care, injury prevention measures, and public health policies, with a focus on addressing the specific needs of pediatric patients in such demanding situations.”

Best regards,

Reviewer 3 Report

The authors have addressed my concerns.

Author Response

(The authors gave the same response as above.)
